# Scanning electron microscopy preparation of the cellular actin cortex: A quantitative comparison between critical point drying and hexamethyldisilazane drying

**Moritz Schu**[1,2], **Emmanuel Terriac**[1], **Marcus Koch**[1], **Stephan Paschke**[3], **Franziska Lautenschläger**[1,2], **Daniel A. D. Flormann**[1,2]*

1 Leibniz Institute for New Materials (INM), Saarland University, Saarbrücken, Saarland, Germany, 2 Center for Biophysics, Saarland University, Saarbrücken, Saarland, Germany, 3 Department of General and Visceral Surgery, University Hospital Ulm, Ulm, Baden-Württemberg, Germany

* Daniel.Flormann@web.de

**Data Availability Statement:** All relevant data are within the manuscript.

## Abstract

The cellular cortex is an approximately 200-nm-thick actin network that lies just beneath the cell membrane. It is responsible for the mechanical properties of cells, and as such, it is involved in many cellular processes, including cell migration and cellular interactions with the environment. To develop a clear view of this dense structure, high-resolution imaging is essential. As one such technique, electron microscopy, involves complex sample preparation procedures. The final drying of these samples has significant influence on potential artifacts, like cell shrinkage and the formation of artifactual holes in the actin cortex. In this study, we compared the three most used final sample drying procedures: critical-point drying (CPD), CPD with lens tissue (CPD-LT), and hexamethyldisilazane drying. We show that both hexamethyldisilazane and CPD-LT lead to fewer artifactual mesh holes within the actin cortex than CPD. Moreover, CPD-LT leads to significant reduction in cell height compared to hexamethyldisilazane and CPD. We conclude that the final drying procedure should be chosen according to the reduction in cell height, and so CPD-LT, or according to the spatial separation of the single layers of the actin cortex, and so hexamethyldisilazane.

## Introduction

The network of the cellular actin cortex assembles just under the cell membrane, and has a typical thickness of 200 nm [1]. It is the main determining factor for the mechanical properties of the cell. Understanding the structure of this thin dynamic meshwork has attracted great interest in recent years [1, 2], especially under conditions where its adaptation is crucial, such as for mitosis [3] and cell migration [4, 5] and differentiation [6].

Given the mesh size of the cellular actin cortex (hole diameter, <100 nm) [1, 2, 7–9], its structural properties are difficult to address with conventional light microscopy. While some recent studies have produced more accurate representations of this network using super-resolution microscopy [10, 11] and rapid atomic force microscopy [8, 12], the main technique

**Funding:** MS, ET, MK, FL, DF: Leibniz Institut für neue Materialien (INM), Saarbruecken, Germany: www.leibniz-inm.de MS, FL, DF: Saarland University Saarbrücken, Germany: www.uni-saarland.de MS, ET, FL, DF: Deutsche Forschungsgemeinschaft [CRC 1027 (A10)]. www.dfg.de SP: Ulm University/University Hospital Ulm, Ulm, Germany: www.uniklinik-ulm.de SP: Deutsche Forschungsgemeinschaft [CRC 1149 (A6)]: www.dfg.de The funders had no role in study design, data collection and analysis, decision to publish, or preparation of the manuscript.

**Competing interests:** The authors have declared that no competing interests exist.

applied in such studies remains scanning electron microscopy (SEM). It is possible to extract quantitative data on the superstructure of the cellular actin cortex from electron micrographs; e.g., the mesh size of the network, and the higher organizational structures, such as bundles of filaments and their patterns of assembly [13].

Most SEM preparation protocols involve the crucial step of sample drying. The preferred method for this step is critical point drying (CPD), which was introduced by Anderson in 1951 [14]. CPD has become an important part of the preparation of biological samples for electron microscopy. The principle of this technique relies on the $CO_2$ phase transition from liquid to vapor, without any further phase transition beyond the critical point [15, 16]. The absence of a potentially damaging phase transition can provide better preservation of the thin super-structures within a sample. Indeed, CPD remains a highly invasive method that can influence the shape and structure of biological samples through the rapid changes in temperature, pressure, and osmolarity, which are also coupled to the relatively high cost of the special equipment needed [17, 18]. As the structure of the cellular actin network is finely regulated by actin nucleators (e.g., formins, Arp 2/3 [19, 20], actin capping proteins [2], and motors, such as myosin), it is important to reduce drying artifacts and discriminate these from real changes that have been induced in the actin cortex structure. Svitkina and co-workers have introduced the addition of lens tissues in-between cover-slips during CPD [21, 22], which does reduce artifacts significantly.

An alternative drying method is the use of hexamethyldisilazane (HMDS). This drying method has a negligible influence on the sample temperature, it is performed under atmospheric pressure, and it has moderate costs, as it is a standard laboratory chemical [18]. In contrast to CPD, the physical background of the mode of action of HMDS drying is poorly understood [23–26]. Empirical studies have shown that HMDS drying can lead to similar results to those obtained with CPD under specific biological conditions [18, 23, 27]. Typically, these are studies of relatively large structures investigating their integrity, or the localization of metallic particles. Consequently, HMDS drying is a widely used method for large structures, while small structures like the cellular actin cortex are commonly investigated using CPD [1, 2, 21]. At the cellular level, both HMDS drying and CPD are known to lead to cell shrinkage [17, 27–32].

Once a structure has been visualized, its quantitative measurements need to be deduced from the images. The reproducibility of the structural analysis of networks has been addressed using different approaches. The classical thresholding and segmentation approaches are widely available. For example, plug-ins such as DiameterJ are implemented in ImageJ, and allow correct tracing if the threshold used reproduces the network sufficiently well, which is only the case if the network differs strongly from the background [33, 34]. For SEM images of the cellular actin cortex, the three-dimensional structures of the actin fibers lead to significant variations in the grayscale values of any single fiber. Therefore, algorithms based on thresholding typically overestimate and/or underestimate the actual network.

To increase the accuracy of the analysis of our imaged networks, we have developed a method for network recognition using a vectoral tracing algorithm in combination with (starting) nodes [35, 36]. Such methods are, for example, based on second-order Gaussian derivatives of each pixel, to follow high or low pixel intensities, so as to recognize fibers correctly. This powerful technique has already been used successfully by Sato and co-workers [37].

Using this technique, we have here compared the preservation of the structural integrity of the cellular actin network after three different SEM drying procedures: HMDS drying, and CPD with lens tissue (CPD-LT) and without lens tissue (CPD).

## Materials and methods

### Cell culture

Immortalized retinal pigmented epithelium (hTERT-RPE1) cells were grown in Dulbecco's modified Eagles medium/F12 medium supplemented with 10% fetal bovine serum (Thermo Fisher, MA, USA), 1% Glutamax (Thermo Fisher, MA, USA), and 1% penicillin/streptomycin (Thermo Fisher, MA, USA) under 5% $CO_2$ at 37˚C, in cell culture flasks (Cellstar, Greiner Bio-One, Austria). For fluorescence imaging, RPE1 cells that stably expressed mCherry LifeAct (kind gift from Matthieu Piel, Paris, France) were used under the same conditions as RPE1 wild-type cells.

### Electron microscopy preparation

**Membrane extraction.**   The SEM samples were generally prepared following the protocols of Svitkina, Chugh, and others [2, 21, 22, 38]. In more detail, the cells were detached from the culture flasks using trypsin, re-suspended in growth medium, and left to adhere onto glass slides (Thermo Fisher Scientific, MA, USA) for at least 24 h. The cells were then rinsed three times with serum-free Leibovitz medium (Thermo Fisher Scientific, MA, USA). To dissolve the cell membranes and to pre-fix the cellular actin network, two treatments were performed. The first membrane extraction and pre-fixation solution was composed of 0.5% Triton X-100, 0.25% glutaraldehyde, and 10 μM phalloidin in buffer M (50 mM imidazole, pH 6.8, 50 mM KCl, 0.5 mM $MgCl_2$, 0.1 mM EDTA, 1 mM EGTA) and was added to the cells for 5 min. The second extraction was composed of 2% Triton X-100 and 1% CHAPS in milliQ water, and was added to the cells for 5 min. The cells were not rinsed between these first and second extractions.

**Cell fixation.**   The cells were fixed by rinsing them three times in buffer M before adding the fixing solution for an overnight incubation, which contained 2% glutaraldehyde (EM grade; Science Service GmbH, Germany) and 2% paraformaldehyde (EM grade: Science Service GmbH, Germany) in 100 mM sodium cacodylate (pH 7.3). Then 0.1% aqueous tannic acid was added for 20 min, without prior rinsing. After the removal of the tannic acid, the cells were washed with distilled water three times. Then 0.2% aqueous uranyl acetate (Science Service GmbH, Germany) was added. After 20 min, the cells were rinsed three times with distilled water. Unless otherwise indicated, all chemicals were purchased from Sigma Aldrich.

**Sample dehydration.**   Following the cell fixation, the samples were ethanol dried using two syringe pumps (with 60-mL syringes). One syringe pump added ethanol, and the second removed the aqueous ethanol solution, to increase the ethanol concentration continuously, thus avoiding discrete steps. In this manner, 60 mL 50% aqueous ethanol was added over 1 h, followed by the same procedure with 100% ethanol. Then, the samples were rinsed twice (without the syringe pumps) with 100% ethanol dried over a molecular sieve, with an incubation time of 20 min after each exchange.

After the HMDS drying and CPD (see below), the samples were sputtered with a layer of 5-6-nm platinum (coater: Model 681; Gatan, USA).

**Hexamethyldisilazane drying protocol.**   After dehydration of the samples through the ethanol, different HMDS drying protocols were tested for the cells, as specified in Table 1. The HMDS drying was performed at room temperature (~23˚C). Therefore, starting from the 100% ethanol dehydration, during the primary HMDS procedure, the HMDS concentration in the 100% ethanol was successively increased according to the step sizes given in Table 1 (i.e., 1%-50%). Each step saw the exchange of all but 1 mL of the previous HMDS concentration in 100% ethanol replaced by the new one, with the next stage incubation according to the

**Table 1. The hexamethyldisilazane (HMDS) drying protocols tested.** The procedures were all carried out at room temperature (~23°C), and completed at the end of the secondary procedures by HMDS evaporation.

| Protocol | Primary HMDS procedure | | | Secondary HMDS procedure | |
|---|---|---|---|---|---|
| Code | HMDS addition step size (%) | Incubation time (min) | Total steps | HMDS wash concentration (%) | Wash time [min (×n)] |
| HMDS1 | 1 | 0.5 | 100 | 100 | 20 (×2) |
| HMDS10 | 10 | 5 | 10 | 100 | 20 (×2) |
| HMDS25 | 25 | 5 | 4 | 100 | 20 (×2) |
| HMDS50 | 50 | 5 | 2 | 98 | 10 (×1) |

incubation times given in Table 1. This process was repeated for the required number of total steps, until a concentration of 98% HMDS (Roth, Austria) was reached. For the further, secondary, HMDS procedure, this 98% HMDS was either exchanged for 99.9% HMDS (Sigma Aldrich, Germany) or kept constant, with washing times generally of 20 min with two washes (Table 1, HMDS1, 10, 25), or for 10 min with one wash for the HMDS50 procedure (Table 1). Finally, the samples were left for the HMDS to evaporate completely.

**Critical point drying protocol.** For the CPD, the fine control of the $CO_2$ exchange rates was initially tested; however, this did not show any significant differences in the results. In the critical point dryer (K850; Quorum Technologies Ltd., UK), the 100% ethanol of the dehydrated samples was exchanged for $CO_2$ following 10 exchange cycles (minimum 50% of the filling level of the chamber). The chamber was then flushed twice with excess $CO_2$ for 20 min per flush, followed by the final temperature increase to 37°C. This was accompanied by the pressure increase to 90 bar. With the temperature kept constant, the pressure was slowly decreased to 1 bar over 40 min. The sample mounting for CPD was realized either with 2.5 mm between each cover-glass (as the 'CPD' protocol), or with lens tissue between each cover-glass (as the 'CPD-LT' protocol).

## Imaging

All of the SEM images were obtained with the scanning electron microscope (SEM Quanta 400 FEG; FEI, USA) using high vacuum mode. For nuclei and cell area measurements, the cells were imaged with a fluorescence microscope (Ti Eclipse; Nikon, Japan) equipped with an incubation chamber (37°C; 5% $CO_2$ in air) for live-cell imaging. The cell nuclei were stained with 50 ng/mL Hoechst. For both fluorescence microscopy and SEM, the cell nuclei and other cell areas were analyzed by thresholding the images in combination with the particle analysis function of ImageJ.

## Atomic force microscopy

The cell height was determined by atomic force microscopy (Nanowizard 3; JPK Instruments, Bruker, USA) with MLCT type C cantilevers (Bruker, USA), using a setpoint force of 2 nN and an approach velocity of 5 μm/s. The cell height measurements were obtained by comparing the contact point on a cell of interest with a contact point on the substrate next to this cell. Quantitative imaging was carried out on dried sputtered samples, with a resolution of 128×128 pixels at 250 μm/s, using the same cantilever and setpoint force as above.

## Classical thresholding image analysis and parameters

Home-made vectorial tracing software was used to characterize the actin network structures imaged by SEM. According to previous studies [1], the mesh hole size is defined by the mesh hole diameter, which for different cell lines is <100 nm. Consequently, a threshold of 110 nm

was used, which included this expected mesh hole diameter plus allowance for the standard deviation. Therefore, every mesh hole diameter >110 nm was assumed to be an artifact that had resulted from the SEM preparation procedures (i.e., a protocol-induced artifactual mesh hole). Additionally, almost all of the mesh holes >110 nm showed broken filaments pointing into the inside of the holes. This supports the definition of these mesh holes as sample preparation artifacts.

The home-made vectorial tracing algorithm is available as open source at https://github.com/SRaent/FiNTA, and it is described elsewhere [39].

## Results

### HMDS10 drying *versus* CPD-LT and CPD

To compare the final drying methods during SEM sample preparation, the hTERT RPE1 cells were prepared as described above. As the preparation methods prior to the final drying stage were the same (i.e., up to the 100% ethanol dried samples), the first steps were carried out without *a-priori* attribution of any particular sample to any particular drying method.

Images of representative cells obtained following each of the three drying protocols for HMDS10, CPD-LT, and CPD are shown in Fig 1, under different magnifications and for

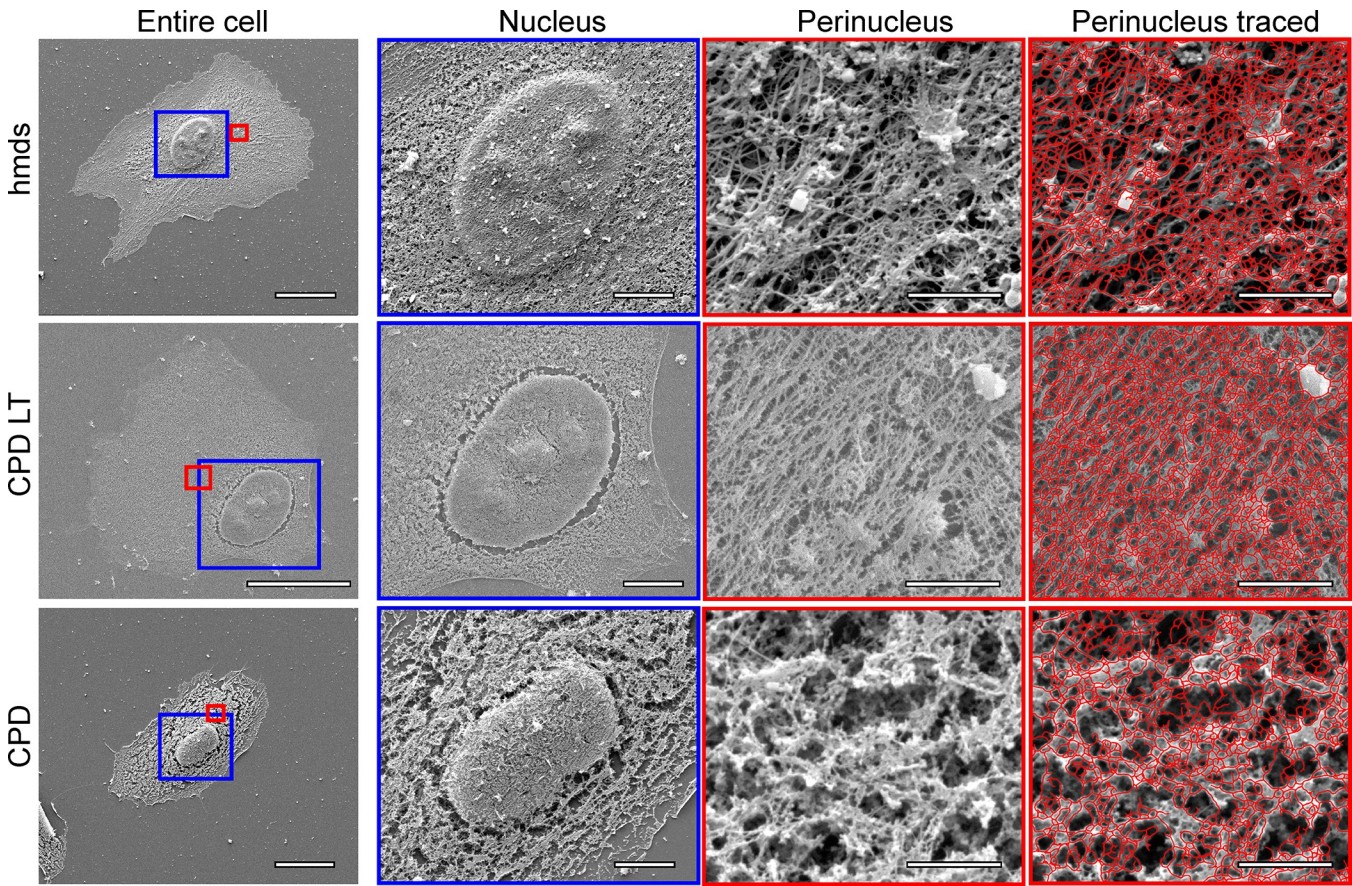

**Fig 1. Representative scanning electron microscopy images of the cells following HMDS10 drying, and CPD-LT and CPD.** Entire cell: whole cell images; magnification, 5k×. Blue boxes, areas defined for nucleus images; red boxes, areas defined for perinucleus images. Nucleus: magnification, 10×. Perinucleus: magnification, 80k×. Perinucleus traced: with software vectorial tracing (red); magnification, 80k×. Scale bars: 20 μm (entire cell); 5 μm (nucleus); 1 μm (perinucleus, perinucleus tracing).

whole cells and different areas of the cells (i.e., over the nucleus, perinucleus). The images with the vectorial tracing by the software are also shown. Of note, the HMDS protocol images in Fig 1 are given for HMDS10 only (see Methods; Table 1), as this provided the most efficient preparation time in combination with the least damage to the cellular actin network of the four HMDS protocols defined (see below and Table 1).

In the images for the 10k× magnification of the nuclei in Fig 1 for the HMDS10 dried and CPD-LT samples, it is already possible to see fewer 'cracks' (holes) in the filamentous network around the nucleus, compared to the CPD samples. The perinuclear images (i.e., the areas between the nucleus and the edges of the cells) under the highest magnification (Fig 1, 80k× magnification) show clear differences between HMDS10 drying, CPD-LT, and CPD that can be seen by eye. Furthermore, comparing CPD-LT to both HMDS10 drying and CPD at this level, the low contrast and the high network density of the CPD-LT images indicate a lower cell height, as investigated further below.

Procedure-induced artifactual mesh holes were defined as holes >110 nm in diameter, as proposed by Chugh and co-workers [1]. The diameters of all of the mesh holes defined by the vectorial tracing software were derived from the measured areas of each mesh hole, on the assumption of circular mesh holes. We used the diameters of these artifactual mesh holes as the measure of the preservation of the structure of the actin cortical network, on the basis that the smaller the artifactual mesh hole diameter, the more preserved the cellular actin network. To quantify these artifactual mesh holes, our vectorial tracing software algorithm was used, as shown in Fig 2. For this method, all mesh holes with diameter <110 nm were ignored in order to focus on artifactual mesh holes only. A representative image of such larger diameter artifactual mesh holes is given in Fig 2A, for the HMDS10 drying protocol.

To fine tune the HMDS protocol for sample preparation, the influence of the exchange rate between ethanol and HMDS was determined as a function of the artifactual mesh hole diameters that were induced by the four HMDS protocols (Table 1). As shown in Fig 2B, there were no significant differences in the artifactual mesh hole diameters created by the HMDS1 or HMDS10 protocols. In contrast, the HMDS25 and HMDS50 protocols led to significantly larger artifactual mesh holes than for the HMDS1 and HMDS10 protocols. Therefore, the HMDS10 protocol was used for all of the further comparisons with CPD-LT and CPD, as it represented the optimum between sample preparation time and quality of the final structure.

The analysis of the artifactual mesh hole diameters for HMDS10, CPD-LT, and CPD are shown in Fig 2C. The artifactual mesh holes for CPD-LT were significantly smaller than for HMDS10. In contrast, CPD led to significantly larger artifactual mesh holes compared to both HMDS10 and CPD-LT. Therefore, we can conclude that CPD is the most error-prone of these methods in terms of the preservation of the cellular actin network through SEM sample preparation. Consequently, for the preparation of such cell samples for SEM analysis, we can recommend the use of HMDS10 or CPD-LT over CPD.

## Correlation between artifactual mesh holes around the nucleus and at the cell perinucleus

The artifactual mesh holes close to the nucleus can be very large compared to those of the cell perinucleus. The artifactual mesh hole diameters within the 2-μm-wide band that followed the outline of the nucleus were compared to those for the perinucleus, as defined as the area between the outer part of the 2-μm band and the cell edge (see Fig 2D and 2E). Fig 2D provides a representative image of the tracing (red) of these artifactual mesh holes with diameter >110 nm around the nucleus after CPD. The diameters of the perinucleus artifactual mesh holes and those of the nuclear band were positively correlated, with Pearson R of 0.11 for HMDS10, 0.33

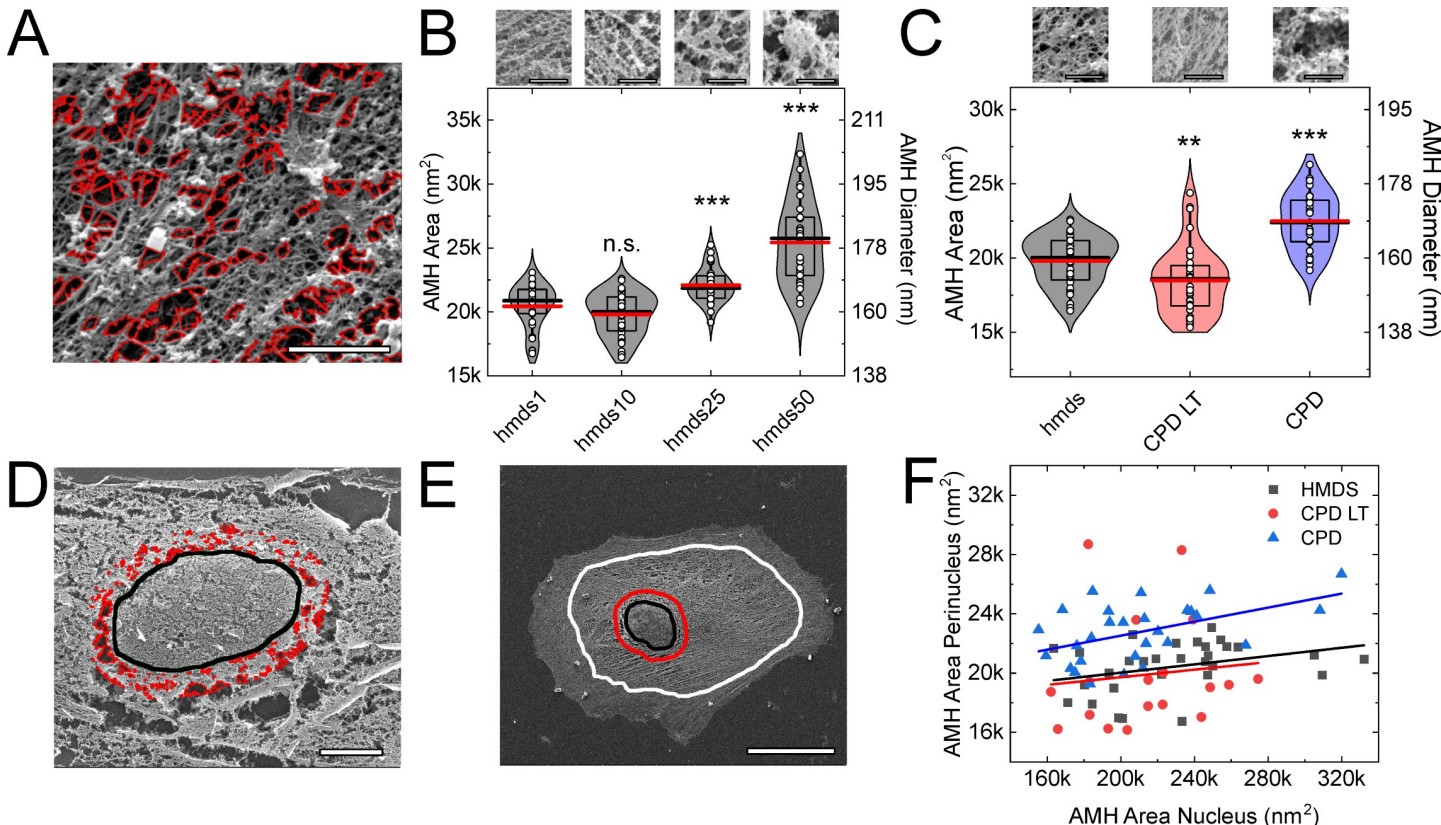

**Fig 2. Artifactual mesh hole analysis.** (**A**) Representative image following HMDS10 drying and vectorial tracing (red) of artifactual mesh holes (magnification, 80k×). (**B**, **C**) Quantification of the artifactual mesh hole sizes according to the different HMDS protocols (Table 1) (B), and for HMDS10 (hmds) in comparison with CPD-LT and CPD as shown in Fig 2 (C). **, $p <0.01$; ***, $p <0.001$; n.s., non-significant; each data set was compared to hmds1 (B) or hmds (C) using t-tests. ANOVA tests: p = $1.88 \cdot 10^{-21}$ (B), p = $5.24 \cdot 10^{-8}$ (C). Scale bars images (B, C): 100 nm. (**D**, **E**) Representative images following CPD showing the vectorial tracing of the artifactual mesh holes (red) within the 2-μm band around the nucleus edge (black line) (D; magnification, 10k×), and the definition of the perinucleus region, shown between the red and white lines (E; magnification, 5k×). (A, D, E) Scales bars: 1 μm (A); 5 μm (D); 10 μm (E). (**F**) Correlation between the artifactual mesh hole areas for the perinucleus and the 2-μm nuclear band following the three SEM preparation protocols (as indicated). AMH, artifactual mesh hole. N = 30, 46, 30, 30 (B, from left to right); N = 46, 37, 28 (C, from left to right); N = 30, 20, 28 (F, hmds, CPD LT, CPD).

for CPD-LT, and 0.49 for CPD (Fig 2F). This correlation between these perinucleus and nucleus artifactual mesh holes can thus be particularly helpful for rapid definition of the quality of SEM samples. Rapid imaging of this nuclear region (i.e., the 2-μm nuclear band) can be enough to determine whether it is worth spending time to analyze any particular sample, in terms of the need to focus on and optimize the parameters for the imaging at the perinucleus.

## Origin of artifactual mesh holes around the nucleus, and cell height

In most cases, artifactual mesh holes were seen around the nucleus, as has been shown in other studies [1, 2, 21, 22, 38]. However, to the best of our knowledge, the origin of these artifactual mesh holes has not been investigated to date. As they are observed in particular around the nucleus, our hypothesis was that they might occur during greater shrinkage of the nucleus, compared to the rest of the cell. Therefore, the nucleus and cell sizes for living and fixed (i.e., 2% paraformaldehyde, 2% glutaraldehyde, in 0.1M cacodylate buffer) cells were measured using fluorescence microscopy, and also following the SEM protocols of HMDS10 drying, CPD-LT, and CPD using SEM (Fig 3A). Additionally, to determine whether cell shrinkage affects the height of the cell, the cell heights above the nucleus and at the cell perinucleus were probed using atomic force microscopy.

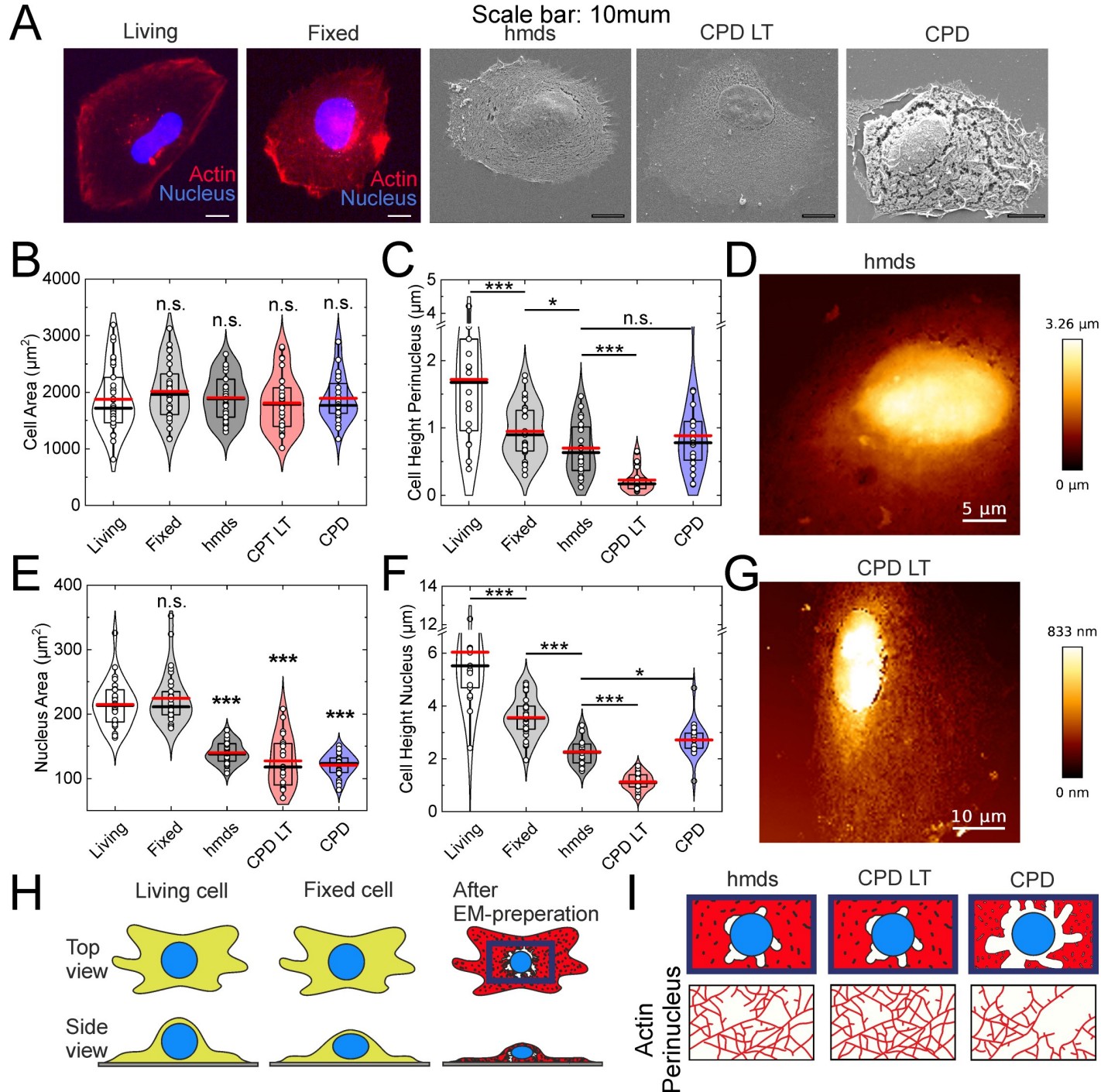

**Fig 3. Influence of cell fixation and SEM drying.** (**A, B, C, E, F**) Representative images (A, scale bars: 10 μm) and quantification from fluorescence microscopy (Living, Fixed) and scanning electron microscopy (hmds, CPD-LT, CPD) of cell (B) and nucleus (E) areas, and from atomic force microscopy (as indicated) of perinucleus (C) and nucleus (F) heights. *, $p < 0.05$; ***, $p < 0.001$; n.s., not significant; unless indicated by lines between conditions all statistical marks compare the conditions to living cells only via t-tests. ANOVA tests: p = 0.58 (B), p = 5.98·10⁻¹¹ (C), p = 5.43·10⁻³⁰ (E), p = 4.9·10⁻²⁴ (F). (**D, G**) Representative atomic force microscopy images of cells following HMDS10 (D) and CPD-LT (G). (**H**) Schematic representations of these three main preparation steps, as living cells, fixed cells, and dried cells (after EM preparation), and (**I**) the resulting appearances of the perinucleus actin network after HMDS10, CPD-LT, and CPD. N = 31, 29, 33, 32, 28 (B, from left to right); N = 31, 29, 30, 20, 27 (E, from left to right); N = 18, 31, 20, 23, 20 (C, from left to right); N = 19, 31, 20, 23, 20 (F, from left to right).

The initial fixation of the cells for the fluorescence microscopy had no significant influence on the total cell areas within the microscopy resolution used, as shown in Fig 3B. In contrast, the nucleus area was significantly reduced for the SEM analysis following HMDS10 drying, CPD-LT, and CPD, compared to living and fixed cells analysed by fluorescence microscopy (Fig 3E). Consequently, we can conclude that the artifactual mesh holes around the nucleus are a result of this significant nucleus shrinkage relative to the unchanged total cell area.

In the cell height analysis for the perinucleus and nucleus, these both showed significant initial decreases from living to fixed cells using fluorescence microscopy (Fig 3C and 3F; by 50%, 40%, respectively). These heights then both showed further significant decreases during the final SEM drying steps for all three of the drying methods (Fig 3C and 3F). Moreover, for both perinucleus and nucleus, the final heights for CPD-LT were half those of both HMDS10 and CPD, which themselves showed similar further reductions in height. Representative atomic force microscopy images are presented in Fig 3D and 3G for only HMDS10 and CPD-LT, as CPD resulted in the same cell heights as for HMDS10. As the cell heights for HMDS10 were double those for CPD-LT, the data from SEM imaging should be used with care. High resolution imaging is the essential feature of SEM, where at the lateral resolution of 1 nm to 10 nm (i.e., range of diameters of single actin fibers), the depth resolution of SEM is from 10 nm to 100 nm (i.e., the SEM focal plane) [40]. At the same time, the thickness of the actin network can vary within several 100 nm [41]. Consequently, SEM imaging of the actin network includes several layers of the actin fibers laid upon each other. Therefore, as CPD-LT results in half the cell height compared to HMDS10, and assuming identical focal planes during the SEM, there might be twice the number of actin layers in the images for CPD-LT compared to HMDS10. Additionally, given the naturally low contrast of thin sample imaging (e.g., Fig 1, CPD-LT), it might not be possible to differentiate between each of several actin layers. In contrast, HMDS10 provided higher contrast in the SEM, and therefore can allow better differentiation between each of several actin layers (e.g., by intensity thresholding). Here, we can conclude that CPD-LT provides information from a slice of actin that was originally thicker than the slice of actin imaged by HMDS10 and CPD. With these all falling within a narrower focal range, this might lead to apparently smaller (i.e., protocol-specific) mesh hole diameters using CPD-LT than for the other two methods here, due to this overlay of several actin layers with CPD-LT. In contrast, HMDS10 (and CPD) will deliver the information from an initially thinner slice of actin, which can therefore enable separation of the actin layers at different heights in the actin network. Thus, while all of these are protocols are expedient for the preparation of the cellular actin network for SEM imaging, their use can depend on the particular interest of a study. Therefore, we recommend CPD-LT as the protocol of choice for investigation of the planar x-y organization of the actin network in cells, or HMDS10 as the protocol of choice for the imaging of potentially single actin layers, due to its superior z-plane integrity.

These fixation and drying effects on the perinucleus and nucleus areas and heights, and their the resulting effects on the cellular actin network, are illustrated schematically in Fig 3H and 3I.

## Nonartifactual mesh holes, network connectivity and fiber length

Up to this point, we have defined the influence of these three SEM protocols on the cellular integrity at the global (i.e., cells, nucleus) and local (i.e., actin network) scale. We then focused on three of the parameters that can be exploited to characterize networks: the 'mesh hole size' after exclusion of the artifactual mesh holes; the network 'connectivity' (also known as the 'coordination number'); and the 'fiber length'. The design and use of this vectoral tracing software is described in [39]. Here, the mesh hole size is defined in the same manner as for the

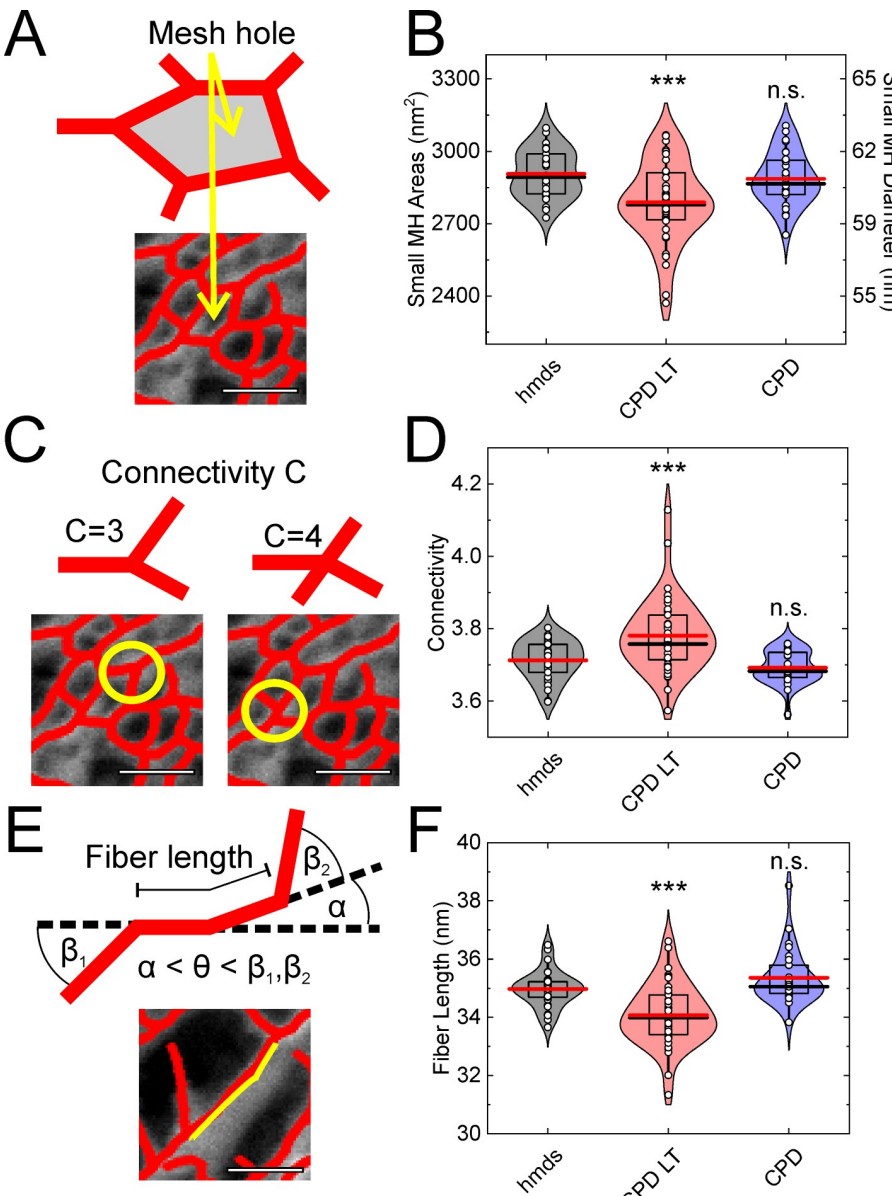

**Fig 4. Mesh hole size, connectivity, and fiber length analysis using the vector tracing software.** (**A**-**C**) Schematic representations of the mesh hole size (A), connectivity (B), and fiber length (C) measures for the cellular actin network. (**D**-**F**) Quantification of the vector tracing software data for HMDS10 drying (hdms), and CPD-LT and CPD, for the mesh hole (MH) size (D; <100 nm; i.e., excluding artifactual mesh holes), connectivity (E), and fiber length (F). Scale bars: 100 nm (A, C, E). *** $p$ <0.001, n.s. not significant; each data set was compared to hmds (B, D, E) using t-tests. ANOVA tests: p = 7.43·10⁻⁴ (B), p = 1.42·10⁻⁵ (D), p = 3.7·10⁻⁷ (F). N = 29, 39, 28 (B, D, F, from left to right).

artifactual mesh hole diameter above (but as <100 nm here) (Fig 4A). The connectivity defines the number of fibers that are connected to a single node (Fig 4B). Finally, the fiber length defines the length of a tracing line up to when the break-off angle (defined here as 17˚; see also [39]) is reached (Fig 4C).

The data from the vectoral tracing software show that all three of these parameters (Fig 4A–4C) are not significantly different between HMDS10 and CPD (Fig 4D–4F). This suggests that although CPD is particularly invasive (i.e., it showed larger artifactual mesh holes [diameter,

>110 nm]; see above), the remaining networks in the cells are similar. In contrast, CPD-LT shows reduced mesh hole size (Fig 4D) and fiber lengths (Fig 4F) compared to both HMDS10 and CPD. Moreover, the connectivity of the actin networks with CPD-LT significantly increased compared to HMDS10 and CPD (Fig 4E). All of these effects should be directly related to the differences in the cell heights shown above.

## Discussion

The production of artifactual mesh holes (diameter, >110 nm) was used here to quantify the quality of SEM preparations of the cellular actin network. Little is known about the precise origin of such artifactual mesh holes, and indeed, there are several explanations possible to define this process. One indicates that they might arise from water or ethanol remaining within samples before the final drying process. During CPD, neither traces of water nor ethanol would reach their critical points (water, 374°C at 221 bar; ethanol, 241°C at 60 bar), and thus the transition from liquid to gas of either of these would be accompanied by a sudden change in density that might lead to fracture of the fine structure within a sample. For HMDS drying, the HMDS itself combines both the low surface tension of ethanol (HMDS, 18 mN/m; ethanol, 22 mN/m) and the low vapor pressure of water (HMDS, 23 mmHg; water, 17.5 mmHg). These properties translate into lower drying rates, and the liquid-gas transition will induce less mechanical stress on the fine microstructure. The potential crosslinking properties of HMDS for biological sample have also been suggested as a reason for the lower shrinkage of biological tissues during drying [26]. However, contrary to CPD, HMDS drying does not involve the critical point, and thus the liquid to gas transition will always be accompanied by changes in density.

In addition, local shrinkage of samples has been shown during HMDS drying and CPD, which can be up to 30% at larger scales, such as with tissue samples [14, 15, 18]. This shrinkage will always lead to local tensions in the samples, which can potentially affect the fine microstructure, such as the actin network.

A further effect might arise from the exchange of solutions during SEM sample preparation. The osmolality of these solutions can vary from 10 mOsmol/L to 50 mOsmol/L (for MilliQ water), to over 300 mOsmol/L (for Leibovitz medium, the first extraction solution), and further to 700 mOsmol/L (for the fixation solution). This study aimed to minimize these effects by the successive and slow addition of each solution, to thus reduce any osmotic shock. However, although the effects of osmotic shock can be reduced in this way, they still cannot be completely excluded.

Finally, CPD has two additional aspects that are not part of HMDS drying. First, the pressure during CPD varies from 1 bar to 90 bar, or even higher. In principle, this should not have a significant influence on the samples as long as there are no air-filled cavities. While we did not note any such cavities in the samples here, the possibility of nanometer-sized gas pockets cannot be fully ruled out. The compression of ethanol during the pressure increase up to 50 bar to 60 bar at the beginning of CPD is in the range of a 0.5% of the volume, and as such, the impact on the sample should be neglectable. However, the degassing of $CO_2$ during the reduction in pressure back to atmospheric pressure is a particularly delicate step. If this happens too quickly, the $CO_2$ can expand and potentially rupture the fine ultrastructure, which will lead to artifactual mesh holes.

## Conclusions

To examine the intact cellular actin network using SEM, sample preparation follows four essential steps: membrane permeabilization, and cell fixation, dehydration, and drying. These

processes can often lead to artifacts, and these appear to arise during the final drying process, which is commonly performed as CPD [1, 2, 21, 22, 38]. HMDS and CPD-LT represent improved alternatives to the classical CPD. Although we have shown here that all three of these drying methods lead to artifacts, the artifactual mesh holes seen for CPD are significantly larger than those for both HMDS and CPD-LT. In addition, CPD-LT leads to approximately half the cell height compared to both HMDS and CPD. Hence, we can conclude that CPD-LT can provide more planar x-y information about the actin network, while also potentially leading to apparently smaller mesh holes. This effect is minimized by HMDS drying, which allows better analysis of single layers of the actin network. Consequently, we conclude that CPD is not suitable for the preparation of samples for analysis of the cellular actin network. In contrast, HMDS drying and CPD-LT provide for more accurate data analysis. Which of these two methods might be preferred will depend on the focus of any investigation.

Moreover, we have shown that the sizes of the artifactual mesh holes around the nucleus and in the cell perinucleus are correlated. Our explanation for this correlation relates to the shrinkage of the nucleus in three dimensions, coupled with little or no change in the area covered by the outer edge of the cell during the sample preparation. We postulate that this collapse of the cellular actin network around the nucleus will lead to increased tension towards the periphery, and therefore the formation of these artifactual mesh holes will serve to reduce this local tension.

## Author Contributions

**Conceptualization:** Moritz Schu, Emmanuel Terriac, Marcus Koch, Franziska Lautenschläger, Daniel A. D. Flormann.

**Data curation:** Moritz Schu, Daniel A. D. Flormann.

**Formal analysis:** Daniel A. D. Flormann.

**Funding acquisition:** Franziska Lautenschläger.

**Investigation:** Daniel A. D. Flormann.

**Methodology:** Moritz Schu, Daniel A. D. Flormann.

**Project administration:** Daniel A. D. Flormann.

**Resources:** Franziska Lautenschläger.

**Software:** Moritz Schu.

**Supervision:** Emmanuel Terriac, Daniel A. D. Flormann.

**Validation:** Daniel A. D. Flormann.

**Visualization:** Daniel A. D. Flormann.

**Writing – original draft:** Daniel A. D. Flormann.

**Writing – review & editing:** Moritz Schu, Emmanuel Terriac, Marcus Koch, Stephan Paschke, Franziska Lautenschläger, Daniel A. D. Flormann.

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
