## [Decision Letter · Decision Letter 0]

5 May 2021

PONE-D-21-01439

Scanning electron microscopy preparation of the cellular actin cortex: a quantitative comparison between critical point drying and hexamethyldisilazane drying

PLOS ONE

Dear Dr. Flormann,

Thank you for submitting your manuscript to PLOS ONE. After careful consideration, we feel that it has merit but does not fully meet PLOS ONE’s publication criteria as it currently stands. Therefore, we invite you to submit a revised version of the manuscript that addresses the points raised during the review process.

We look forward to receiving your revised manuscript.

Kind regards,

Kanhaiya Singh, Ph.D

Academic Editor

PLOS ONE

Additional Editor Comments:

Although the reviewers have found this study interesting, they are suggesting inclusion of comparison of cryo-preparations. Reviewers are also suggesting to revisit the statistical tests applied in this paper.

Journal Requirements:

Reviewers' comments:

Reviewer's Responses to Questions

**Comments to the Author**

1. Is the manuscript technically sound, and do the data support the conclusions?

Reviewer #1: Yes

Reviewer #2: Yes

Reviewer #3: Partly

2. Has the statistical analysis been performed appropriately and rigorously? 

Reviewer #1: Yes

Reviewer #2: Yes

Reviewer #3: No

3. Have the authors made all data underlying the findings in their manuscript fully available?

Reviewer #1: Yes

Reviewer #2: Yes

Reviewer #3: No

4. Is the manuscript presented in an intelligible fashion and written in standard English?

Reviewer #1: Yes

Reviewer #2: Yes

Reviewer #3: Yes

5. Review Comments to the Author

Reviewer #1: The authors present a comparative study about different preparation (drying) methods of brain cortex for scanning electron microscopy. They compare standard criyical point drying (CPD), with CDP plus lens tissue (CDP-LT), and hexamethyldisilazane drying, of which the last method created the least number of artifacts (hole, etc.). Effects of the methods on cell thickness have been monitored by atomic force microscopy. This is a concise paper, and all the results seem flawless and appropriately executed. This may be a relevant study for all these locations that cannot resort to any cryo methods, and where the low-resolution results shown here still bear enough data for a scientific comparison or analysis.

The issue I have with this paper is more of a philosophical nature, than a technical one. The gold-standard for preparations today of cells and cellular components should always be cryo-SEM (or cryo-EM in general), even if this is not used, e.g., in a high-throughput approach, or in a clinical environment, where results have to be achieved rapidly. Cryo-EM or cryo-SEM warrants the least interference with distortions, and as such, all other methods should be compared to this. In this study, all methods used, no matter how standardized and commonly used they are, produce large amounts of distortions and alterations to the cellular fine structures. Hence, I would consider cryo-methods the gold standard, to which everything else should be compared to (especially for the atomic-force height measurements). That will not reduce the relevance of the techniques tested here, but it would place them into a more realistic perspective of what all this drying is actually doing to a biological sample, no matter which method is chosen.

To make this paper acceptable for me, I would like to see a comparison to cryo-preparations. There are serious artifacts from all three preparations that render an objective comparison quite difficult. I believe the results shown here, but I am not sure that one method (HMDS) is truly superior and should allow more sophisticated conclusions than the other two. Hence, I recommend acceptance after major revisions (adding cryo-data). Cryo-SEM is quite common these days, especially in Germany. There are multiple institutions that could help with that.

Reviewer #2: Manuscript is done, the authors clearly discuss the pro and cons of different drying methods for Electron microscopy.

I would like the authors to address the figure 4, Please include the magnification, scale bar on the SEM images.

Reviewer #3: 1. In Fig 2B, please show the images for HMDS 1, HMDS25 and HMDS50 so that its easy to compare them. Please use Anova to test the significance as there are more than 2 groups.

2. Is Fig.2C quantification for Fig 1? Please clarify.

3. Fig 3FA, B, D,E please show the representative fluorescence and SEM images. Please specify "n" number in the legend. Use Anova to test significance.

4. In Fig 3, please show the atomic absorption images of living, fixed and CPD.

5. In Fig. 4, please show representative images for B, D, F. Please use anova to test the significance since comparison is between 3 groups.

6. PLOS authors have the option to publish the peer review history of their article (what does this mean?). If published, this will include your full peer review and any attached files.

Reviewer #1: No

Reviewer #2: No

Reviewer #3: No

---

## [Author Response · Author response to Decision Letter 0]

25 May 2021

The responds to reviewers may be found in the uploaded file "Response to reviewers"

---

## [Decision Letter · Decision Letter 1]

22 Jun 2021

Scanning electron microscopy preparation of the cellular actin cortex: a quantitative comparison between critical point drying and hexamethyldisilazane drying

PONE-D-21-01439R1

Dear Dr. Flormann,

We’re pleased to inform you that your manuscript has been judged scientifically suitable for publication and will be formally accepted for publication once it meets all outstanding technical requirements.

Kind regards,

Kanhaiya Singh, Ph.D

Academic Editor

PLOS ONE

Additional Editor Comments (optional):

Reviewers' comments:

Reviewer's Responses to Questions

**Comments to the Author**

1. If the authors have adequately addressed your comments raised in a previous round of review and you feel that this manuscript is now acceptable for publication, you may indicate that here to bypass the “Comments to the Author” section, enter your conflict of interest statement in the “Confidential to Editor” section, and submit your "Accept" recommendation.

Reviewer #2: (No Response)

2. Is the manuscript technically sound, and do the data support the conclusions?

Reviewer #2: (No Response)

3. Has the statistical analysis been performed appropriately and rigorously? 

Reviewer #2: (No Response)

4. Have the authors made all data underlying the findings in their manuscript fully available?

Reviewer #2: (No Response)

5. Is the manuscript presented in an intelligible fashion and written in standard English?

Reviewer #2: (No Response)

6. Review Comments to the Author

Reviewer #2: (No Response)

7. PLOS authors have the option to publish the peer review history of their article (what does this mean?). If published, this will include your full peer review and any attached files.

Reviewer #2: **Yes: **imrankhan

---

## [Editor Report · Acceptance letter]

28 Jun 2021

PONE-D-21-01439R1 

Scanning electron microscopy preparation of the cellular actin cortex: a quantitative comparison between critical point drying and hexamethyldisilazane drying 

Dear Dr. Flormann:

I'm pleased to inform you that your manuscript has been deemed suitable for publication in PLOS ONE. Congratulations! Your manuscript is now with our production department. 

Kind regards, 

on behalf of

Dr. Kanhaiya Singh 

Academic Editor

PLOS ONE